# Prophylactic Administration of Tranexamic Acid Reduces Blood Products’ Transfusion and Intensive Care Admission in Women Undergoing High-Risk Cesarean Sections

**DOI:** 10.3390/jcm12165253

**Published:** 2023-08-12

**Authors:** Yair Binyamin, Amit Frenkel, Igor Gruzman, Sofia Lerman, Yoav Bichovsky, Alexander Zlotnik, Michael Y. Stav, Offer Erez, Sharon Orbach-Zinger

**Affiliations:** 1Department of Anesthesiology, Soroka University Medical Center, The Faculty of Health Sciences, Ben-Gurion University of the Negev, Beer-Sheva 84101, Israel; igorgr@clalit.org.il (I.G.); alekszl@clalit.org.il (A.Z.); 2General Intensive Care Department, Soroka University Medical Center, The Faculty of Health Sciences, Ben-Gurion University of the Negev, Beer-Sheva 84101, Israel; frenkela@clalit.org.il (A.F.); bichovsk@gmail.com (Y.B.); 3Department of Anesthesia, Beilinson Hospital, Rabin Medical Center Associated with Sakler Medical School, Tel Aviv University, Tel Aviv 6423906, Israel; michaelstav86@gmail.com (M.Y.S.); sharonorbach@yahoo.com (S.O.-Z.); 4Division of Obstetrics and Gynecology, Soroka University Medical Center, The Faculty of Health Sciences, Ben-Gurion University of the Negev, Beer-Sheva 84101, Israel; offererez@gmail.com

**Keywords:** tranexamic acid, cesarean section, post-partum hemorrhage, prevention, hemoglobin, hysterectomy, renal failure, prophylaxis

## Abstract

Postpartum hemorrhage (PPH) remains a major cause of maternal mortality. Tranexamic acid (TxA) has shown effectiveness in reducing PPH-related maternal bleeding events and deaths. We conducted a cohort study including parturient women at high risk of bleeding after undergoing a cesarean section (CS). Participants were divided into two groups: the treatment group received prophylactic 1-g TxA before surgery (n = 500), while the comparison group underwent CS without TxA treatment (n = 500). The primary outcome measured increased maternal blood loss following CS, defined as more than a 10% drop in hemoglobin concentration within 24 h post-CS and/or a drop of ≥2 g/dL in maternal hemoglobin concentration. Secondary outcomes included PPH indicators, ICU admission, hospital stay, TxA complications, and neonatal data. TxA administration significantly reduced hemoglobin decrease by more than 10%: there was a 35.4% decrease in the TxA group vs. a 59.4% decrease in the non-TxA group, *p* < 0.0001 and hemoglobin decreased by ≥2 g/dL (11.4% in the TxA group vs. 25.2% in non-TxA group, *p* < 0.0001), reduced packed red blood cell transfusion (*p* = 0.0174), and resulted in lower ICU admission rates (*p* = 0.034) and shorter hospitalization (*p* < 0.0001). Complication rates and neonatal outcomes did not differ significantly. In conclusion, prophylactic TxA administration during high-risk CS may effectively reduce blood loss, providing a potential intervention to improve maternal outcomes.

## 1. Introduction

During delivery, there is an activation of the fibrinolytic system, seen in the increased plasma urokinase-plasminogen activator (u-PA) or tissue plasminogen activator (t-PA) levels and decreased plasminogen activator inhibitor 1 (PAI-1) [1]. Previous studies have shown that tranexamic acid (TxA) significantly reduced maternal death due to postpartum hemorrhage (PPH) and reduced bleeding in cesarean and other surgeries [2,3,4,5,6,7,8]. A randomized control trial provide evidence of TXA reducing estimated blood loss (but not blood loss based on gravimetric assessment) in patients undergoing cesarean delivery [9].

Because of this evidence, its low cost, and the fact that it does not require refrigeration, the World Health Organization (WHO) classified it as an essential medicine, especially in low-resource countries [10].

Because of its effectiveness in reducing the risk of death from PPH, there is growing interest in its use as prophylaxis for prevention of PPH during cesarean sections (CS) and delivery [9,11,12,13,14].

Although there is controversy regarding this latter point, most articles conclude that there is no role for prophylactic TXA given in all CS. However, most of these studies excluded women at high risk of bleeding [9,11].

We hypothesized that these parturients may be the ones that would most benefit from prophylactic TxA and, therefore, we decided to conduct an exploratory cohort impact study to determine the prophylactic effect of TxA administration during CS of women at high risk of obstetrical hemorrhage (multiple pregnancy, third cesarean section and above, macrosomia, placental abruption, placenta previa with bleeding, abnormal placentation disorders, PET with severe characteristics, HELLP syndrome, and eclampsia) on hemoglobin, other parameters of PPH, ICU admission and length of hospital stay, TXA complications, and neonatal outcome.

## 2. Materials and Methods

Study Design: The study was a pragmatic retrospective impact study, a secondary analysis of our previous study examining all CS [15].

Ethics: The hospital institutional review board approved the study and waived patient informed consent as this was a retrospective study (approval number 0054-20-SORof Soroka University Medical Center, Beer Sheva, Israel).

Eligibility: Starting in 2017, all women undergoing CS received prophylactic TXA. We collected data on 1000 high-risk women who underwent low segment CS, of which 500 received 1 g prophylactic TxA and 500 did not. CS at high risk of bleeding was defined as: multiple pregnancy, third cesarean section and above, macrosomia, placental abruption, placenta previa with bleeding, abnormal placentation disorders, PET with severe characteristics, HELLP syndrome, and eclampsia [16,17]. Exclusion criteria were intrauterine fetal demise.

Recruitment: Two time frames were analyzed: (1) the control group: women who had CS before the introduction of the routine administration of TxA (1 January 2014 until 30 September 2014), and (2) the study group: women who had prophylactic administration of TxA before CS (1 January 2019 until 14 September 2019). In order to ensure compliance; a five-year gap between the periods was chosen since the introduction of prophylactic TXA was carried out in a gradual manner.

Setting: Soroka University Medical Center (SUMC) is the largest maternal ward in the country, with over 17,000 deliveries yearly. It is a tertiary hospital located in the south of Israel.

Procedure: Study women received 1 g of TxA before skin incision, given over 30–60 s.

Organization: There was no change in anesthetic or surgical technique between the two time frames.

Follow up: Women were followed up for three months using the hospital’s computerized medical records.

Primary outcome: (1) incidence of women having a hemoglobin drop >10% 24 h after CS compared to pre-operative baseline and (2) incidence of women having a hemoglobin drop of ≥2 g/dL. 

Secondary outcome: maternal outcome (hemoglobin change, estimated blood loss as assessed by surgeon, need for blood products (packed red blood cell, fresh frozen plasma, cryoprecipitate and platelets, cesarean hysterectomy) (blood products were given in case of a hemoglobin level below 7.0 or during active bleeding with clinical manifestations or when symptoms of anemia appeared), intensive care admission, length of hospital stay), TxA complications (acute renal failure, increase in creatinine, thromboembolism, seizures) and neonatal data (Apgar, pH). Follow up of complications was for a three-month period because of thromboembolic events being more common in the postpartum period until at least 12 weeks after delivery [18].

Statistical Analysis: Continuous parametric data are presented with their mean ± standard deviation and comparisons performed using the independent group’s Student’s t-test. Categorical data are presented as count (percentage) and comparisons are performed using the Chi-square test. The data were analyzed and processed using R software (version 4.0.0). Because there were two primary outcomes, a Bonferroni correction was made and *p* < 0.025 was considered significant for the primary outcome. As the secondary outcomes were exploratory, they were not corrected.

A multivariable logistic regression model was performed to assess the primary outcome hemoglobin decrease >10% while controlling for maternal age, BMI, parity, number of CS, pregnancy week and TxA administration.

Sample size calculation: The study by Lakshmi and Araham [19] reported that 40% of parturients that did not receive tranexamic acid had a drop of more than 10% in hemoglobin level 24 h after surgery compared to its level before surgery. We assumed that a risk reduction of 25% in the proportion of women with a hemoglobin drop of more than 10% to be a clinically significant difference. Based on these assumptions and a significance level (alpha) of 0.05 and a power of 90%, the number of parturients that needed to be included was 475 per group. Taking into account that data were lacking or insufficient, 5% of our samples, a minimum of 500 parturients per group was necessary.

## 3. Results

The clinical and demographic characteristics of the study groups are presented in Table 1. The two groups had no differences in demographic variable or baseline obstetric data. 

Primary outcome: The rate of hemoglobin decrease by >10% was higher in the non-TxA vs. the TxA group (Non-TxA group 59.4% (297/500) vs. TxA group 35.4% (177/500), *p* < 0.0001). The rate of hemoglobin decrease by ≥2 gm/dL was higher in the non-TxA group versus the TxA group (non-TxA 25.2% (126/500) vs. TxA11.4% (57/500), *p* < 0.0001) (Table 2). 

Secondary outcomes other indicators of PPH: The prophylactic administration of TxA was associated with a reduced delta hemoglobin (pre vs. post CS hemoglobin), estimated blood loss, packed red blood cell transfusion within 48 h of CS. There was no difference in need for hysterectomy or need for intraoperative blood transfusion (Table 2).

Secondary outcomes: ICU admission and hospital stay were shorter in the group that received TxA (Table 2).

Secondary outcomes: Complications. The groups had no difference in the rate of acute renal failure, delta creatinine concentrations, thromboembolic events, or seizures (Table 3). All patients who had seizures had a history of preeclampsia.

Secondary outcomes: Neonatal data. Neonatal outcomes are listed in Table 4. There were no significant between-group differences in the incidence of any neonatal outcome.

Multivariable regression: Using a multivariable logistic regression model where we controlled for maternal age, BMI, parity, number of CS, gestational age at surgery, anemia before surgery, and TxA administration, administration of prophylactic TxA and number of CS were independently associated with lower rates of hemoglobin drop >10% (TxA administration—β = 0.787, 95% CI 0.7410; 0.8359, *p* < 0.0001. number of CS—β = 0.9694, 95% CI 0.9462; 0.9933, *p* = 0.0126) (Table 5).

Using another multivariable logistic regression model controlling for maternal age, BMI, parity, number of CS, gestational age at surgery, anemia before surgery, and TxA administration, administration of prophylactic TxA was not significantly associated with rates of visual estimated blood loss >1000 mL (TxA administration-β = 0.778, 95% CI 0.5055; 1.1922, *p* = 0.2513) (Table 5).

## 4. Discussion

In this analysis of real-world data, we found that the use of TxA in high-risk CS decreased blood loss and the need for postoperative packed red blood cell transfusions, as well as decreasing ICU admissions and shortening hospitalization time. Using a multivariable logistic model controlling for other factors influencing PPH, we found that TxA administration was independently associated with decreased blood loss.

Although studies showing that the use of prophylactic TXA in all CS had questionable benefits [9,11,20,21,22], we believe that our study does provide evidence for its use prophylactically in high-risk women.

Different studies have used different outcome measures to assess bleeding. In the study by Senthilles, hemorrhage was defined as calculated estimated blood loss > 1000cc or administration of RBC infusions within two days of delivery [9]. Because we included women with preoperative anemia, we felt that it was more accurate to use an absolute measure, a decrease in hemoglobin of >2 gm or a percentage decrease greater than 10%. 

In terms of secondary outcomes, we report here for the first time that the prophylactic administration of TxA reduces the rate of ICU admission in women at risk of obstetrical hemorrhage who undergo CS. This was observed in our study population, which was included in previous reports [2,9]. Moreover, an additional novel observation by our study is the shorter hospitalization among women who received prophylactic TxA treatment. This parameter was not measured in previous reports [9], but has a great impact on public health expenses and patients’ satisfaction. 

In recent years, paradoxically, the overall rate of PPH in high-income countries has increased; however, it was not associated with a concomitant elevation in maternal mortality [23]. There are many possible factors that can result in maternal mortality, including increasing maternal age, the increased incidence of multiple gestation, higher rates of placenta previa and placenta accreta due to previous CS [23], and a higher rate of intrapartum CS. Most RCTs have been limited to low-risk pregnancies, and thus the specific dynamics of parturients expected to bleed have not been studied; thus, the importance of real-world data in examining PPH cannot be overemphasized.

Two studies are currently being conducted investigating the use of prophylactic TxA in other high-risk populations, including women with prepartum anemia and placenta previa [24,25]. A recent secondary analysis of a TRAAP2 study by Sentilhes suggests that TXA was not associated with a reduction in any hemorrhage-related outcome among women with multiple pregnancies [26].

We also investigated the effect of prophylactic TXA administration prior to skin incision rather than following the delivery of the neonate [9]. We are not able to discuss the neonatal outcome but rest assured that there is no difference in pH or Apgar score.

Our study did not show that TxA caused unexpected side effects. In a recent article examining World Health Organization database case safety reports, there were 29 cases of adverse renal vascular and ischemic complications, 42 reports of pulmonary embolisms, and 41 reports of peripheral embolism and thrombosis in women receiving TxA [27]. In this study, however, 52.9% of women received doses higher than 1gm. Likewise, in a French study describing 18 cases of renal cortical necrosis after postpartum hemorrhage, high dose TxA was used including a bolus of 1–4 gm followed by a maintenance infusion [28]. This finding is in contrast with the WOMAN’s trial [2], which included 20,000 women and did not find that the use of TXA was associated with the risks of renal failure, embolism, or seizures. The incidence of renal failure, seizures, or thromboembolic events in our study was 0.2%, 0.6%, and 0.2%, respectively; however, our study did not aim primarily to assess these outcomes. We used hemoglobin change rather than EBL as our primary measure of outcome. We decided to use hemoglobin change because EBL has been shown to underestimate blood loss, especially at high volumes. However, hemoglobin can also be influenced by other factors, including hemodilution.

The strengths of our study are our large numbers and the strict adherence to protocol. In Senthilles’ study, ¼ of women deviated from protocol [9]. In our study, as the use of TxA was part of a strictly protocolized regimen, we examined treatment as opposed to intention to treat.

Our study’s limitations include the retrospective data, which make some information from charts inaccessible. Second, the historical nature of the study may cause the difference in results to be secondary to the changes in practices, such as intrauterine tamponade balloon. Third, we have no information on whether the surgery was conducted during labor or before labor. Fourth, our cohort has a high gravida and parity and a very high prior CS history, which may make our generalizability limited. Fifth, numerators and incidence of complications were extremely low, especially ICU admission. Therefore, our study is unable to ascertain whether TxA is associated with a reduced risk of ICU admission or RBC transfusion. Unfortunately, we do not have data on patients who received a repeat dose of TxA; a recent article showed the utility of a repeat dose in cases of continued fibrinolytic activity [29].

## 5. Conclusions

Prophylactic TxA administration may reduce estimated blood loss and hemoglobin changes among women at high risk of obstetrical hemorrhage undergoing cesarean delivery in our cohort. Further high quality randomized controlled studies must be performed to confirm this finding.

## Figures and Tables

**Table 1 jcm-12-05253-t001:** Characteristics of the participants.

Characteristic	Tranexamic Acid Group (N = 500)	No Tranexamic Acid Group (N = 500)	*p* Value
Age (years)	32.30 ± 5.73	32.88 ± 5.47	0.0987
Height (cm)	161.3 ± 5.7	161.7 ± 5.8	0.2157
Weight (Kg)	81.05 ± 15.28	80.65 ± 13.89	0.6587
Gravity—median (range)	4 (1–25)	4 (1–16)	0.583
Parity—median (range)	3 (0–12)	3 (0–12)	0.5865
Previous cesarean delivery—no. (%)	339 (67.8%)	338 (67.6%)	0.9442
Number of cesarean delivery—median (range)	3 (1–8)	3 (1–8)	0.8572
Gestational diabetes—no. (%)	54 (10.8%)	56 (11.2%)	0.8414
HTN chronic—no. (%)	13 (2.6%)	13 (2.6%)	1.000
Hypothyroidism—no. (%)	21 (4.2%)	18 (3.6%)	0.6241
Kidney disease—no. (%)	- (0.0%)	- (0.0%)	-
History of PPD—no. (%)	8 (1.6%)	9 (1.8%)	0.8103
Anticoagulation Therapy	32 (6.4%)	23 (4.6%)	0.2113
Pregnancy week	36.75 ± 2.55	36.56 ± 2.88	0.2697
Singletons	406 (81.2%)	415 (83%)	0.4593
Twins	88 (17.6%)	81 (16.2%)	0.5552
Triplets	6 (1.2%)	3 (0.6%)	0.3173
Quadruplets	0 (0%)	1 (0.2%)	0.3173
Placenta previa	26 (5.2%)	19 (3.8%)	0.2846
Placenta accreta	6 (1.2%)	3 (0.6%)	0.3173

PPD—Postpartum depression.

**Table 2 jcm-12-05253-t002:** Maternal outcomes.

Outcome	Tranexamic Acid Group (N = 500)	No Tranexamic Acid Group (N = 500)	*p* Value
Delta hemoglobin (gr/dL)	0.85 ± 1.02	1.42 ± 1.01	<0.0001
Hemoglobin decrease ≥ 2 gr/dL—no (%)	57 (11.4%)	126 (25.2%)	<0.0001
Hemoglobin drop >10%—no (%)	177(35.4%)	297 (59.4%)	<0.0001
Estimated blood loss% (mL)	705.2 ± 178.06	766.82 ± 172.48.26	<0.0001
Estimated blood loss > 1000 mL—no (%)	42 (8.4%)	53 (10.6%)	0.2808
Emergent hysterectomy—no (%)	3 (0.6%)	3 (0.6%)	1
Packed red blood cell transfusion during surgery—no (%)	7 (1.4%)	15 (3%)	0.1313
Packed red blood cell transfusion during surgery above 3 units	1 (0.2%)	3 (0.6%)	0.3173
FFP transfusion during surgery—no (%)	5 (1%)	9 (1.8%)	0.4194
Cryoprecipitate transfusion during surgery—no (%)	5 (1%)	6 (1.2%)	1
Platelet transfusion during surgery—no (%)	2 (0.4%)	6 (1.2%)	0.2869
Packed red blood cell transfusion 48 h—no (%)	21 (4.2%)	40 (8%)	0.0174
FFP transfusion 48 h—no (%)	5 (1%)	6 (1.2%)	1
Cryoprecipitate transfusion 48 h—no (%)	4 (0.8%)	4(0.8%)	1
Platelet cell transfusion 48 h—no (%)	3 (0.6%)	4 (0.8%)	1
ICU admission—no (%)	2 (0.4%)	9 (1.8%)	0.034
Hospital stay (days)	4.52 ± 1.48	5.12 ± 2.4	<0.0001
Hospital stay > 5 days—no (%)	59 (11.8%)	109 (21.8%)	<0.0001

**Table 3 jcm-12-05253-t003:** Complications.

Complication	Tranexamic Acid Group	No Tranexamic Acid Group	*p* Value
Acute renal failure no (%)	1 (0.2%)	3 (0.6%)	0.6164
Delta creatinine (mg/dL)	−0.003 ± 0.11	−0.006 ± 0.02	0.9276
Thromboembolic event no (%)	1 (0.2%) [DVT − 1]	2 (0.4%) [DVT − 1, PE − 1]	1
Seizure no. (%)	3 (0.6%)	8 (1.6%)	0.2252

**Table 4 jcm-12-05253-t004:** Neonate outcomes.

Outcome	Tranexamic Acid Group N = 592	No Tranexamic Acid Group N = 581	*p* Value
APGAR 1 min—median (range)	9 (0–9)	9 (0–9)	0.1724
APGAR 1 min < 5—no (%)	23 (3.89%)	27 (4.64%)	0.5157
APGAR 5 min—median (range)	10 (0–10)	10 (0–10)	0.7422
APGAR 5 min < 5—no (%)	11 (1.86%)	15 (2.58%)	0.4009
PH	7.29 ± 0.082	7.28 ± 0.093	0.3314
PH < 7.2—no (%)	35 (5.91%)	50 (8.6%)	0.07508

**Table 5 jcm-12-05253-t005:** Multivariable logistic regression model for the association between prophylactic tranexamic acid and hemoglobin drop > 10% and for the association between prophylactic tranexamic acid and visual EBL > 1000 mL.

**Multivariable Logistic Regression Model for the Association between Prophylactic Tranexamic Acid and Hemoglobin Drop >10%**			
	**Odds Ratio**	**95% CI**	***p*** **Value**
Age	0.99414	0.9681; 1.0207	0.6629
BMI	1.00449	0.9784; 1.0312	0.7376
Parity	1.06255	0.9936; 1.1370	0.0771
Number of CS	0.88044	0.7917; 0.9781	0.0180
Pregnancy week	1.0033	0.9558; 1.0531	0.8931
TxA administration	0.3807	0.2931; 0.4932	<0.0001
Anemia (Hb < 11.0)	0.5626	0.4290; 0.7361	<0.0001
**Multivariable Logistic Regression Model for the Association between Prophylactic Tranexamic Acid and EBL > 1000 mL**			
	Odds Ratio	**95% CI**	***p*** **Value**
Age	0.9848	0.9414; 1.0296	0.5037
BMI	0.9940	0.9498; 1.0379	0.7919
Parity	1.1157	1.0063; 1.2323	0.0332
Number of CS	1.017393	0.8648; 1.1963	0.8345
Pregnancy week	0.9748	0.9063; 1.0549	0.5091
TxA administration	0.7690	0.4986; 1.1792	0.2303
Anemia (Hb < 11.0)	1.1929	0.7698; 1.8365	0.4250

EBL—estimated blood loss.

## Data Availability

yairben1@gmail.com.

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
