# Peer review of "Prophylactic Administration of Tranexamic Acid Reduces Blood Products’ Transfusion and Intensive Care Admission in Women Undergoing High-Risk Cesarean Sections"

_jcm, 2023, doi:10.3390/jcm12165253_

Round 1

Reviewer 1 Report

This article focused on a meaningful clinical issue based on the analysis of real-world pragmatic data. It provide information for the TXA use in the high-risk CS. There are two suggestions for modifying.

1. In view of the influence of the surgical team, surgical method, hemostatic method and techniques on the hemorrhagic outcomes, please supplement the corresponding information in the two groups and make comparison between two groups to determine whether the decrease in blood loss and delta hemogloubin attribute to the improvement of surgical techniques. And these confounders should be included in the multivariable analysis.

  • 2. Anemia is proved to be a risk of PPH, and the proportion of anemia should be provided in the two groups. The perioperative level of hemogloubin should also be included in the multivariable analysis. 

Minor editing of English language required

Reviewer 2 Report

It is an interesting study, despite being retrospective with its limits well noted at the end of the article, it concerns pregnant women.

In the title of the article, it is noted Tranexemic acid, in the text it is Tranexamic.

To be able to further support the beneficial effect of Tranexamic Acid, I would be even more interested to know if the authors can, if possible, complete it taking into account the following remarks:

About Tranexamic Acid drug:

-          1 g used, corresponds to the standard dosage per injection, taking into account the weight of the patients. It is useful to note this in the text: i.e. a single injection on, average of 12.3 mg/Kg (i.e. 10.3 mg/Kg to 15 mg/Kg).

-          1g injected over 30 to 60 seconds before skin incision: the 1g is in 10 ml, normally it must be injected in 1 ml/minute intravenously, i.e. over 10 minutes; unless you have another packaging or other mode of administration. On the other hand, indeed, the time of action is fast and it is coherent to inject it in pre-incision of the skin. We consider that, according to the defined protocol, that a single dose was used per patient over the entire period of hospitalization, it did not provide for a new injection within 24 hours of caesarean section, even in patients who bled.

• In the two groups of patients, among the parameters at risk of bleeding collected, it is not specified whether or not there are additional non-obstetric parameters but at risk of bleeding:   -          The presence or absence of a clinical hemorrhagic phenotype in everyday life, before caesarean section (therefore predisposing to bleeding in a surgical situation?   -          The presence or absence of a biological phenotype, for example a disorder of primary hemostasis or coagulation in everyday life (therefore predisposing to bleeding in a surgical situation)?

-          Presence of a "non-physiological" biological blood abnormality at the end of pregnancy (predisposing to bleeding in a surgical situation).

-          Is information available for each patient on possible nonsteroidal anti-inflammatory or antidepressant intake (IRS-N family) or anti-thrombotic treatment, at the end of pregnancy or even very close to Cesarean (predisposing to bleeding in a surgical situation)?

-          Was the initial Hb level measured overall at what time in relation to the start of the caesarean section? Was there a necessary pre-cesarean hemodilution in some patients?

-          In the primary endpoint: when was the Hb level measured after cesarean section? Did some patients, in both groups, benefit from hemodilution / hydration after caesarean section?

-          Has drug thromboprophylaxis been implemented in all caesarean patients? It was started before or after H24 of the caesarean section?

-          On what Hb level you relied for the post-caesarean RBC transfusion (this is the differential ≥ 2 g/dL or a predefined threshold and/or on the clinical symptoms).

-          What does the Cryoprecipitates administered mean for you: concentrates of Factor VIII or Factor Willebrand (the fibrinogen concentrate is for us a cryo-supernatant)?

Renal tolerance was good, a reassuring result (even expected) under this single dose of Tranexamic Acid (non-cumulative).

Finally, in the case of a prospective study, it will be demonstrated that the use of a second injection of Tranexamic Acid within the following 24 hours (given its half-life) could further reduce the risk of bleeding or the bleeding itself in patients at risk.

Important Work.  

Best regards.
